# How to Make Sense out of 75,000 Mesenchymal Stromal Cell Publications?

**DOI:** 10.3390/cells11091419

**Published:** 2022-04-22

**Authors:** Dina Mönch, Marlies E. J. Reinders, Marc H. Dahlke, Martin J. Hoogduijn

**Affiliations:** 1Dr. Margarete Fischer-Bosch Institute of Clinical Pharmacology, 70376 Stuttgart, Germany; dina.moench@ikp-stuttgart.de; 2University of Tübingen, 72074 Tübingen, Germany; 3Erasmus MC Transplant Institute, Department of Internal Medicine, Erasmus University Medical Center, 3015 GD Rotterdam, The Netherlands; m.e.j.reinders@erasmusmc.nl; 4Department of Surgery, Robert-Bosch-Hospital, 70376 Stuttgart, Germany; marc.dahlke@rbk.de

**Keywords:** mesenchymal stromal cell, cell therapy, animal models, clinical trials

## Abstract

Mesenchymal stromal cells have been the subject of an expanding number of studies over the past decades. Today, over 75,000 publications are available that shine light on the biological properties and therapeutic effects of these versatile cells in numerous pre-clinical models and early-phase clinical trials. The massive number of papers makes it hard for researchers to comprehend the whole field, and furthermore, they give the impression that mesenchymal stromal cells are wonder cells that are curative for any condition. It is becoming increasingly difficult to dissect how and for what conditions mesenchymal stromal cells exhibit true and reproducible therapeutic effects. This article tries to address the question how to make sense of 75,000, and still counting, publications on mesenchymal stromal cells.

## 1. Introduction

Since their first description in the 1970s by Friedenstein et al. [1,2,3], researchers have embarked en masse on studying the function and therapeutic potential of mesenchymal stromal cells (MSC). Especially the discovery in the late 1990s that MSC possess stem cell properties [4] triggered a steep rise in the number of MSC-related publications. Pubmed.gov (accessed on 18 February 2022) displays 167 publications on MSC in the year 2000 and this number went up to more than 8000 in 2021. Currently, every day, more than 20 publications appear, and in total, more than 75,000 publications on MSC have been published so far. Many of these publications demonstrate therapeutic effects of MSC in diverse preclinical and in vitro models. The FDA currently lists over 1400 interventional clinical trials, of which over 350 early trials (phase 1 or phase 2) have been completed and over 80 have been enrolled in phase 3 (www.clinicaltrial.gov (accessed on 18 February 2022), search term “mesenchymal stem or stromal cell”). With so many publications and clinical trials, it is unavoidable that conflicting results are reported, and that supporting literature can be found for diverse hypotheses on the mechanisms of action and therapeutic effects of MSC. The overwhelming number of publications makes it difficult to filter out solid data that can bring the field further in developing MSC-based tools and real clinical therapies. The question is how researchers can make sense out of such a massive amount of data.

## 2. Therapeutic Effects of Mesenchymal Stromal Cells

MSC combine regenerative and immunomodulatory properties, and therefore, represent an attractive therapeutic option for a highly diverse group of degenerative and immunological disorders. They can be harvested from different tissues that are either considered renewable or discarded, with bone marrow and adipose tissue representing the traditionally most commonly used sources, and newer sources such as neonatal tissues (umbilical cord, placenta, amniotic fluid, amniotic membrane) [5]. MSC possess a multilineage differentiation potential and their ability to differentiate into adipocytic, chondrocytic, or osteocytic cells [4] laid the foundation for the initial studies on MSC as a therapeutic option for bone or cartilage repair and improvement of bone marrow transplantation in patients with hematologic malignancy [6,7]. Clinicians need to decide whether to use autologous or allogeneic MSC for their studies, each of which have their own advantages and disadvantages. Autologous MSC are easy to obtain and do not evoke allo-immune responses, but it requires time to isolate and in vitro expand MSC on an individual basis [8]. Moreover, the anticipated use of autologous MSC may be expensive, because each culture has to undergo its own extensive safety testing prior to infusion [5]. Concern was also raised that the underlying disease might affect the performance of MSC. Studies in systemic lupus erythematosus (SLE) reported a growth retardation in vitro and a decreased secretion of some cytokines in MSC from SLE patients compared to healthy donors [9]. However, these findings might be limited to specific disease settings. BM-MSC from patients with systemic sclerosis exhibited the same properties as their healthy counterparts [10,11] and comparison of autologous MSC from patients with end-stage renal disease and healthy individuals revealed phenotypical and functional similarities, which indicated that they might be equally suitable for MSC therapy [12]. Allogeneic MSC, on the other hand, provide a convenient “off the shelf” solution, with a growing selection of different donors from various sources, and always sufficient cell numbers [8]. Especially the interaction with the recipients’ immune system is a highly discussed topic with regards to when to decide whether to use allogeneic or autologous MSC, as there are animal studies reporting anti-donor immune responses with repeated allogenic, but not autologous MSC injections [13], and studies that report a transition from a so-called “immunoprivileged” to an immunogenic state after differentiation [14]. Clinical trials directly comparing allogeneic versus autologous MSC application are rare and inconclusive. The POSEIDON trials, which compared allogenic vs. autologous MSC in patients with different types of cardiomyopathies, demonstrated favorable safety profiles for both cell types, but did not see significant superior effects of one cell type [15,16]. In patients with type 1 diabetes, allogenic and autologous MSC were applied in combination with hematopoietic stem cells (HSC) and the combination of autologous MSC and HSC resulted in better long-term control of hyperglycemia than the allogenic group [17]. Thus, both applications bear their own risks and limitations, but also advantages.

The knowledge gain on MSC functional characteristics that we have seen over the years has broadened the spectrum of possible therapeutic applications. The original idea of using MSC to replace damaged cells has, thus, been subjoined by the discovery of their ability to secrete cytokines and growth factors, which stimulate native tissue repair pathways and modulate immune responses of the recipient. As a result of this, studies investigating the effect of MSC exist for almost every indication: regenerative medicine, such as cartilage or tissue repair of the musculoskeletal and nervous system [18]; organ transplantation and the prevention of graft-versus-host disease (GvHD) [19,20,21]; ischemic injuries [22]; diabetes [23]; autoimmune diseases and chronic inflammatory disorders, such as multiple sclerosis; inflammatory bowel diseases and chronic obstructive pulmonary disease (COPD) [24,25,26]; and even psychiatric disorders, such as autism [27].

The preclinical studies in the abovementioned fields reported predominantly positive outcomes. Preclinical MSC therapies were, hence, able to demonstrate safety and efficacy in a variety of diseases, such as colitis [28,29], wound healing [30], cardiovascular diseases [31,32], arthritis [33], and organ transplantation [34,35,36,37], among others. Meta-analysis on various preclinical models demonstrated that the efficacy of MSC therapy was maintained across different animal models, MSC origin, source, and route of administration [38,39,40,41,42]. This is in conclusion with our own comprehensive survey in animal models of inflammatory diseases and other studies, which reported therapeutic effects regardless of alterations in the MSC product [43,44,45]. To allow for a reliable comparison of MSC efficacy between studies, detailed information on MSC characterization, culture conditions and administration need to be available, which is not always the case [38,43]. Thus, future research should include detailed methodological descriptions to facilitate the comparison and evaluation of different animal models. Finally, it is important to keep in mind that a publication bias towards successful studies is not unlikely, and that informative data from studies that demonstrate a lack of effects of MSC in particular disease models and under particular conditions is not available via published works.

The promising preclinical animal data encouraged researchers to test MSC therapy in human clinical trials. Phase I/II clinical trials conducted in kidney [46,47,48], liver [49,50,51], lung [52,53,54,55,56,57], and small bowel disease [58,59,60] confirmed the safety and feasibility profile, but the outcomes of the advanced clinical trials fell short of the research community’s expectations regarding the efficacy of MSC treatment [61,62]. It needs to be clearly considered that more than 20 years lie between the first test of MSC as a cellular therapeutic in humans performed by Lazarus in 1995 [7] and the first clinical phase III trials that showed statistically significant therapeutic effects. Among these, Alofisel achieved clinical remission of treatment-refractory, complex perianal fistulas in patients with Crohn’s disease for up to 104 weeks [63,64]. Remestemcel-L treatment showed an improved response rate in pediatric patients who failed to respond to steroid treatment for acute GvHD [65]. Clearly, phase III trials are extremely time and money consuming, and corporate involvement is essential for such large endeavors. Currently, there is no strong market rewarding MSC therapies, at least not compared to the development of small molecule therapies. It will be essential for the future of MSC therapies that this changes.

Recently, the SARS-CoV-2 pandemic has developed into a novel playground for the exploration of therapeutic effects of MSC in inflammatory lung diseases; since March 2020, over 400 publications appeared studying MSC therapy for COVID-19. SARS-CoV-2 induces an acute release of cytokines that lead to increased vascular permeability, pulmonary edema, vessel congestion, and impairment of air exchange across the membranes. In severe cases, this may lead to ARDS and death [66]. The MSC-based therapeutic approach aims to abort or minimize this cytokine storm by MSC infusion, thereby reducing lung damage and promoting the restoration of tissue function through inherent reparative properties [66]. Several clinical trials indicated that MSC therapy indeed attenuates the cytokine storm in patients with severe COVID-19 [67,68,69,70,71]. These studies report a significant decrease in inflammatory cytokines (IL-6), an accelerated recovery of damaged lung tissue [71], and an increase in anti-inflammatory cytokines, especially in patients with initially high pro-inflammatory cytokine levels, suggesting patients with high IL-6 levels or patients in the acute inflammatory phase might be more likely to benefit from MSC treatment [68,71]. However, the small sample size, the lack of appropriate control groups, and the inclusion of patients who were also receiving concomitant drugs, limit the significance of these data and makes the requirement of further investigations essential.

Thus, taken together, possible explanations for the discrepancy between preclinical animal studies and human clinical trials regarding the efficacy of MSC therapies are diverse and may best be summarized by a lack of immune compatibility, variable dosing and timing regimens, and the overall fitness of cultured MSC [72]. For example, it was demonstrated that the timing of MSC infusion for immunomodulation is of special importance in patients receiving kidney transplantation. Pre-transplant MSC infusion provided a safety advantage over post-transplant MSC infusion at day seven, as an impaired graft function was observed, when MSC were administered at day seven after kidney transplantation, but not when administered before transplantation [73]. In addition, we still have limited knowledge on the inflammatory microenvironment in which MSC are deployed [74]. In vitro studies showed that changes in viscoelasticity and mechanotransduction altered the surface structures of the surrounding microenvironment, and in turn, the cytokine profile of MSC [75,76]. In a clinical setting, an altered cytokine secretome could well impair the efficacy of MSC therapies. The major challenge for future research will be to close the gap between the efficacy of animal models and insufficient outcome of human clinical trials by addressing these topics and by installing standardized protocols.

## 3. Standardization and Harmonization of MSC Therapy

In the last decade, a great diversification in MSC products, treatment indications, and delivery methods has occurred [77]. This resulted in a lack of standardization and harmonization of MSC manufacturing protocols as well as a variation in cell culture processing. While BM-MSC still seems to be the gold standard, MSC are also isolated from adipose tissue and perinatal tissues, which could affect the therapeutic potential.

Indeed, the investigation of MSC donor and tissue heterogeneity is a highly discussed research area with most contradictory results. Many studies outlined that MSC from different tissues show differences in growth kinetics, differentiation abilities, immunophenotype, proteome, secretome, and immunomodulative abilities [78,79,80,81]. However, it was speculated that aspects such as isolation efficacy, frequency, and expansion may more likely affect MSC clinical exploitation [82]. A comparative analysis of BM-MSC and adipose tissue-derived MSC from the same donor also showed that both MSC types had similar immunomodulatory capacity with only minor differences in the potency to inhibit different leukocyte subsets [83]. Additionally, there are also reports on donor-to-donor variation in MSC of the same tissue origin, suggesting that donor age or health might influence MSC function or availability [84,85]. Different proliferation rates and differentiation capacities were also found in healthy donors of the same age [86]. Recently, it was shown that aging and expansion of in vitro cultured BM-MSC, rather than in vivo donor aging, changed the characteristics of MSC and induced epigenetic changes that could impact their therapeutic outcomes [87,88]. Extensively passaged human MSC undergo morphological, phenotypic, and genetic changes, which are also found to be modulated by the medium composition employed to expand the cells [89]. Additionally, whole gene expression profiles (especially cell surface markers) of in vitro cultured MSC differ in between freshly and longer cultured MSC [90].

The reason that many studies are not only difficult to reproduce but difficult to evaluate for comparability, and the impact within the field might also be attributed to the fact that heterogeneity is rarely investigated in individual research projects and the unique cell populations used within these studies might not be fully defined [91]. Moreover, compared to the regulatory guidelines that apply for traditional pharmaceuticals, the regulatory requirements for MSC therapies in terms of quality assurance and quality control are, at best, obsolete [91]. For a cellular therapeutic product such as MSC, potency assays may be more informative than phenotypic details. The challenge will be to set up guidelines that consider safety and efficacy and find a compromise between the required quality controls and economic and therapeutic efficacy.

## 4. Mechanisms of Action of Mesenchymal Stromal Cells

Thousands of publications focus on the mechanisms of action of MSC. The paradigms through which MSC are believed to exercise their therapeutic effects has changed considerably over the years [43]. Initially, it was thought that MSC would engraft after administration and display long-term effects from their site of engraftment in tissues. Deeper investigation into the bio-distribution and longevity of MSC revealed that MSC have problems passing narrow capillary networks due to their size, and that they have a short intravascular survival time [31,92]. These results led to the currently prevailing hypothesis that the therapeutic effects of MSC are mediated via their secretome. The MSC secretome is rich in cytokines, chemokines, and growth factors and micro-environmental conditions can modify the abundance of these factors [93]. A plethora of studies in diverse in vitro and in vivo disease models demonstrate that MSC mediate angiogenic processes through their paracrine function [94], and that a broad range of secreted factors mediate the immunomodulatory and organ regenerative effects of MSC in immune and injury models [28,95,96]. Currently, over 7000 publications investigating or reviewing the immunomodulatory actions of MSC are currently listed on pubmed, and although the cellular and molecular mechanisms have not been fully clarified, there are certain pathways that have been repeatedly indicated to play a role. MSCs operate through a combination of direct cell-cell-contact and soluble factors on many types of immune cells including dendritic cells (DC), monocytes/macrophages, B/regulatory B cells (Breg), T/regulatory T cells (Treg), Th1/Th2 and Th17 helper cells, natural killer (NK)/natural killer T (NKT) cells, innate lymphoid cells (ILC), myeloid-derived suppressor cells (MDSC), neutrophils, and mast cells [97,98]. Depending on the context or disease, the effects seen by MSC application either promote or suppress the immune response. Furthermore, a number of studies indicate that the viability of MSC is not a prerequisite for their immunomodulatory effects and that some of the immunomodulatory effects of MSC can be mediated by apoptotic, metabolically inactivated, or fragmented MSC [98,99,100,101].

An important component of the MSC secretome are extracellular vesicles (EV). The last decade has seen a steep rise in the number of studies on MSC-derived EV, and today, over 3500 publications are available. MSC-derived EV contain extracellular matrix proteins and cell adhesion proteins [102] and over 300 types of microRNAs [96], through which they may induce immunomodulatory and regenerative processes [103]. As an example, MSC-derived extracellular vesicles have been demonstrated to mediate regenerative processes in acute kidney injury [104] and ischemia-induced liver injury [105]. Their ability to pass capillary networks and their non-replicative nature, and therefore, excellent safety profile, have evoked major interest in studies to the biological and therapeutic effects of extracellular vesicles across different fields.

While the secretome hypothesis can certainly explain the effects of MSC in certain models, the question is whether the potent effects of MSC after intravenous infusion can be rhymed with the secretion of relevant amounts of soluble factors and extracellular vesicles during the brief presence of MSC in a significantly large bodily fluid. An alternative explanation could be that MSC instruct host cells to change their behavior, and in turn, modulate biological processes. It has been demonstrated that MSC are induced to undergo apoptosis after infusion and are subsequently engulfed by phagocytic cells [106]. These phagocytic cells subsequently adapt their phenotype and function and modulate biological processes even after the MSC have disappeared [107,108].

The above studies indicate that insights on the mechanism of action of MSC have changed over the years. The development of novel, better, hypotheses will help the development of MSC-based therapies.

## 5. How Should We Proceed?

Although the vast majority of preclinical studies demonstrated therapeutic effects of MSC, most of the clinical studies proved safety, but were not able to demonstrate efficacy due to small patient numbers and natural variability within study populations. Therefore, the successful translation to clinical efficacy has been, and still is, a huge challenge. Strategies to enhance the efficacy of clinical trials may include patient stratification to identify disease phenotypes that might best respond to MSC therapy, better knowledge of the inflammatory microenvironment, MSC licensing, and a more accurate and adjusted definition of endpoints in clinical trials [74,109]. Additionally, timing of MSC treatment and the concurrent immunosuppressive medication may be instrumental in achieving therapeutic efficacy of MSC [110].

Apart from differences between humans and animals or varying treatment regimens, insufficient efficacy in clinical trials could also be caused by an underreporting of negative-outcome studies in preclinical models. Because of the lack of such publications, we get the impression that MSCs work for everything and proceed to clinical testing without a solid fundament. Negative scientific results in general are often disregarded or neglected and not forwarded for publication in peer-reviewed journals. A study from 2012 showed that there is a trend to even more “positivity” in scientific literature [111], which enhances the lack of reproducibility of scientific results [112]. Although researchers highly value publication of “negative” results, they often do not publish their own, citing lack of time and the perception that those results may not be as highly cited as reasons [113]. In addition, only 11 to 49% of the published studies can be reproduced [114]. In terms of MSC-based therapies, these problems could be addressed by stricter acceptance criteria for MSC studies for journals or a special search mode, in which unreproducible data appear at the bottom. This should then let relevant and true results that can bring the field further come to the surface.

A decade ago, Hartshorne and Schachner proposed the tracking of reproducibility as a method of post-publication evaluation similar to the citation count and also presented a theoretical set-up for this, which unfortunately stayed at the theoretical set-up level [115]. In the end, it will have to be the common effort of MSC scientists, editors, and publishers, as well as funding agencies, to embrace an honest discussion of negative results and acknowledge their impact on the progress of the field.

## 6. Where Will MSC Research Eventually Lead to?

The ever-expanding MSC library is filled with useful studies to the biological and therapeutic effects of MSC, but also with contradictory studies and one-off reports. Publications on MSC span all corners of the medicine field across different eras. The richness in MSC publications makes it complicated to filter out findings that are relevant for the development of therapy for a particular indication. While the number of publications is still rising sharply, clinical studies that are able to demonstrate efficacy have been limited. The question is whether the wave of publications hints towards the dawn of the therapeutic era of MSC [43], or whether they merely reflect research activity and are not necessarily indicative for clinical breakthroughs. The coming years will likely see many more publications added to the library, and some of them will be instrumental to determine whether or not MSCs have therapeutic effects in particular conditions. When this has been settled, the field may focus on a selective number of promising indications, which may be followed by a welcome decline in the number of publications.

## 7. Conclusions

The celebration of 75,000 publications on MSC means that there is an overwhelming amount of data that are impossible to read, even for a large collective of researchers. Especially for new researchers in the field, the large number of publications is a hurdle to becoming familiar with all published data, and on top of all these publications, they may be missing a large number of unpublished negative studies that would be helpful to form a balanced view of the therapeutic possibilities and impossibilities of MSC.

In addition to the use of impact factors as a quality indicator for published studies, a reproducibility indicator would be helpful in filtering out data that are solid enough to use as building blocks for the design of rational clinical studies with MSC. This could help selecting indications for which MSC may be expected to have therapeutic benefit, and reconsider testing MSC for indications for which such evidence does not exist.

A focused follow-up of successful studies and abandoning of non-reproducible and negative studies may move the field forward. The “sense” of 75,000 MSC publications is that we have information on all these issues. We just need to filter out useful information for specific clinical purposes.

## Data Availability

Not applicable.

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
