# Peer review of "How to Make Sense out of 75,000 Mesenchymal Stromal Cell Publications?"

_cells, 2022, doi:10.3390/cells11091419_

Round 1

Reviewer 1 Report

The review articles presented by Mönch and colleagues is really interesting. the amount of papers regarding MSC therapeutic potential is huge and sometimes controversial. 

For this reason, my opinion is that this review summarizes very well the problem that we face in the literature.

I would only suggest the authors to add 3 tables to summarize the paper cited in the chapter:

Therapeutic effects of mesenchymal stromal cell

Standardization and harmonization of MSC therapy

Mechanisms of action of mesenchymal stromal cells 

This will help the reader to better follow the topics.

Author Response

Dear Editors and reviewers,

Please find enclosed our answer to the reviewers’ comments on our manuscript “How to make sense out of 75,000 mesenchymal stromal cell publications?”.

We found the reviewers‘ comments particularly knowledgeable and would like to address these as follows:

Reviewer 1 suggested to add three tables to summarize the papers cited in the chapters “Therapeutic effects of mesenchymal stromal cells”, “Standardization and harmonization of MSC therapy” and “Mechanisms of action of mesenchymal stromal cells” to better help readers following the different topics.

The reviewer addresses a very good point, however, in the final version of our manuscript, after careful consideration, we decided against including summarizing tables. The main goal of the manuscript it to address the problems that arise from the large amount of literature about mesenchymal stem cells. Thus, the literature we refer to in the chapters is not intended to be exhaustive but rather aims to accentuate the overwhelming amount and diversity of available literature. We feel the insertion of tables for an overview makes a biased selection of the 75000 publications, which is in our opinion against the message of the manuscript.  

It is our hope that you can accept the manuscript for publication in Cells, and that we will hear from you in due course.

Yours sincerely,

Dina Monch

Reviewer 2 Report

This report touches one very import issue regarding the enormous number of publication on mesenchymal stem cells and their clinical usefulness.

 The analysis was properly done and the  related presentation of the results useful to any reader interested on this issue.

Many question marks are raised and proper responses are given

It  deserves to be published. 

Author Response

Dear Editors and reviewers,

We thank reviewer 2 for their time to read our work and for the positive comments to our manuscript. We hope our manuscript can attract interest by researchers in the field.
Best regards,
the authors.